

# Assessing the prognostic scores for the prediction of the mortality of patients with acute-on-chronic liver failure: a retrospective study

Yue Zhang*, Yuan Nie*, Linxiang Liu and Xuan Zhu

Department of Gastroenterology, The First Affiliated Hospital of Nanchang University, Nanchang, Jiangxi, China
* These authors contributed equally to this work.

Corresponding author
Xuan Zhu, waiyongtg@163.com

## ABSTRACT

**Background:** Acute-on-chronic liver failure (ACLF), which is characterized by rapid deterioration of liver function and multiorgan failure, has high mortality. This study was designed to identify prognostic scores to predict short-term and long-term outcome in patients with ACLF to facilitate early treatment and thereby improve patient survival.

**Materials and Methods:** We retrospectively analyzed 102 ACLF patients who were hospitalized in the gastroenterology department. The EASL-CLIF criteria were used to define the ACLF. The demographic characteristics and biochemical examination results of the patients were acquired, and seven scores (CTP score, MELD score, MELD-Na, CLIF ACLF score, CLIF-C OF score, and CLIF SOFA score) were calculated 24 h after admission. All patients were observed until loss to follow-up, death, or specific follow-up times (28 days, 3 months, and 6 months), which were calculated after the initial hospital admission. The receiver operating characteristic (ROC) curve was employed to estimate the power of six scores to forecast ACLF patients' outcome.

**Results:** All scores were distinctly higher in nonsurviving patients than in surviving patients and had predictive value for outcome in patients with ACLF at all time points ($P < 0.050$). The areas under the ROC curve (AUROCs) of the CLIF-SOFA score were higher than those of other scores at all time points. The comparison of the AUROC of the CLIF-SOFA score with other scores was statistically significant at 28 days ($P < 0.050$), which was the only time point at which it was greater than 0.800.

**Conclusion:** Patients with ACLF have high mortality. These six scores are effective tools for assessing the prognosis of ACLF patients. The CLIF-SOFA score is especially effective for evaluating 28-day mortality.

## INTRODUCTION

Acute-on-chronic liver failure (ACLF) is a clinical syndrome characterized by the rapid deterioration of liver function due to acute injury. Patients diagnosed with ACLF often

have multiple organ failures and high short-term mortality (*Bernal et al., 2015*). Patients with chronic liver disease may progress to liver failure induced by enhanced viral replication, combined with bacterial or fungal infection and liver injury due to drug abuse or alcoholism (*Biggins et al., 2018*). The basic etiology of ACLF is mainly alcoholism in European and American countries; however, hepatitis virus infection is the leading etiology of ACLF in Asian countries, especially in China (*Zhao et al., 2018*). Although treatments such as liver transplantation and hemodialysis markedly improve survival in the short term, they are not extensively obtainable in clinical practice because of their high costs, the limited availability of liver resources, and the need for hospitalization. ACLF causes a heavy economic burden on patients. ACLF patients perform obvious differences in accordance with morbidity and survival. So, it is essential to develop an applicable prognostic score to estimate the outcomes in ACLF patients and help guide doctors in determining the treatment options according to the predicted outcomes.

Some prognostic scores have been established previously. The Child-Turcotte-Pugh (CTP) score was first established as a widely utilized liver-specific score nearly 50 years ago (*Pugh et al., 1973*). Wiesner's research analyzed data and established the Model for End-Stage Liver Disease (MELD) score; the MELD score is superior to the CTP score with regard to the prediction of 3-month mortality in patients with chronic end-stage liver disease (*Wiesner et al., 2003*). The MELD combined with serum sodium concentration (MELD-Na) score is related to the MELD score and has improved prognostic efficacy in cirrhotic patients awaiting liver transplantation (*Kim et al., 2008*). In the EASL-CLIF ACLF in cirrhosis (CANONIC) study, ACLF was defined using a novel scoring system called the CLIF-sequential organ failure assessment score (CLIF-C SOFA), which is a modification of the original SOFA score. The EASL-CLIF consortium also developed the CLIF consortium organ failure score (CLIF-C OF), which simplified the original CLIF-SOFA. Through further studies, *Jalan et al. (2014)* found that age and white blood cell (WBC) count were independent risk factors for mortality and established the CLIF-C ACLF score. The CLIF-C ACLF score not only assesses the effects of extrahepatic organ injury, coagulation and circulatory failure but also includes age and inflammatory indicators; the CLIF-C ACLF score has high clinical value for evaluating the prognosis of ACLF. Up to now, there are less study on comparing all methods for the evaluation and prediction of prognosis in ACLF patients with a variety of etiologies, especially among Asians. Our study was designed to assess the short-term and long-term discriminative power of all of the above scores in ACLF patients to direct clinical practice.

## MATERIALS AND METHODS

### Study patients

Our study was a single-center retrospective study that was completed in ACLF patients hospitalized in our institute between January 2015 and December 2018. Patients were included when they fulfilled these criteria: (a) ≥18 years old and (b) diagnosed with cirrhosis and ACLF (defined by the EASL-CLIF Consortium). Exclusion criteria included (1) hepatocellular carcinoma, (2) previous liver transplantation, (3) complications with other severe chronic extrahepatic diseases and (4) infection with human

immunodeficiency virus. Our study was approved by the Ethics Committee of the First Affiliated Hospital of Nanchang University (No. 2015-1203). All the patients signed the informed consent.

## Definitions

Cirrhosis was defined by laboratory tests, radiologic imaging, endoscopy or liver biopsy. The ACLF criteria and organ failures were defined based on the CLIF-SOFA score according to the EASL-CLIF Consortium. The ACLF grading system classifies patients with ACLF in 1 of 3 grades according to the number of organ failures as per the CLIF-SOFA score as follows: Grade 1 if (1) single kidney failure (serum creatinine level ≥2.0 mg/dl) or (2) another organ failure (respiration, circulation, coagulation, or liver) is accompanied by grade I–II (West Haven criteria) hepatic encephalopathy (HE) and/or a serum creatinine level of 1.5–1.9 mg/dl, or (3) single cerebral failure (grade III–IV HE) is present with a serum creatinine level of 1.5–1.9 mg/dl; grade 2 if 2 organ failures are identified; or grade 3 if 3 or more organ failures have been diagnosed. The Child-Pugh score was computed based on albumin, ascites, HE, prothrombin time (PT), and serum bilirubin (*Pugh et al., 1973*). The MELD formula was: $3.8 \times \log$ (bilirubin) $+ 9.6 \times \log$ (creatinine) $+ 11.2 \times \log$ (INR) $+ 6.43$ (*Kamath et al., 2001*). The MELD-Na score was calculated as below: MELD–Na $= (0.025 \times \text{MELD} \times (140–\text{Na})) + 140$ (*Kim et al., 2008*). The CLIF-SOFA score was computed as the sum of the scores for six organ systems, including the cardiovascular, hepatic, coagulation, respiratory, nervous, and renal systems (*Moreau et al., 2013*). The CLIF-C OF score includes the revised six organ systems of the CLIF-SOFA score. The CLIF-C ACLF score was revised according to the CLIF-SOFA score and was computed with the formula: $10 \times (0.63 \times \log$ (white-cell count) $+ 0.33 \times$ CLIF-C OF $+ 0.04 \times$ age$–2)$ (*Jalan et al., 2014*).

## Study protocols

Patients with ACLF were included in the study. During hospitalization, data were collected regarding medical records, demographics, the presence of other comorbidities, clinical features, the number of complications and type of decompensation, the etiology of cirrhosis, and blood haematological index at admission (such as blood platelet count, WBC count, the INR, renal function test, liver function test). The patients were followed up for 6 months to obtain survival information. Patients with incomplete follow-up at 28 days, 3 months, and 6 months were not included in the final analysis of the corresponding time.

## Statistical analysis

The statistical analyses were performed using SPSS software version 20.0 (SPSS Inc., Chicago, IL, USA). Continuous variables were expressed as the mean ± standard deviation (SD) or medians (interquartile range (IQR)), and categorical data were expressed as percentages. Differences in variables were analyzed using Student *t*-tests or the Mann–Whitney U test. Categorical variables are described as the frequencies (percentages (%)) and were compared with chi-squared or Fisher's exact tests. Receiver operating characteristic (ROC) curves were used to measure the performance of the score for the

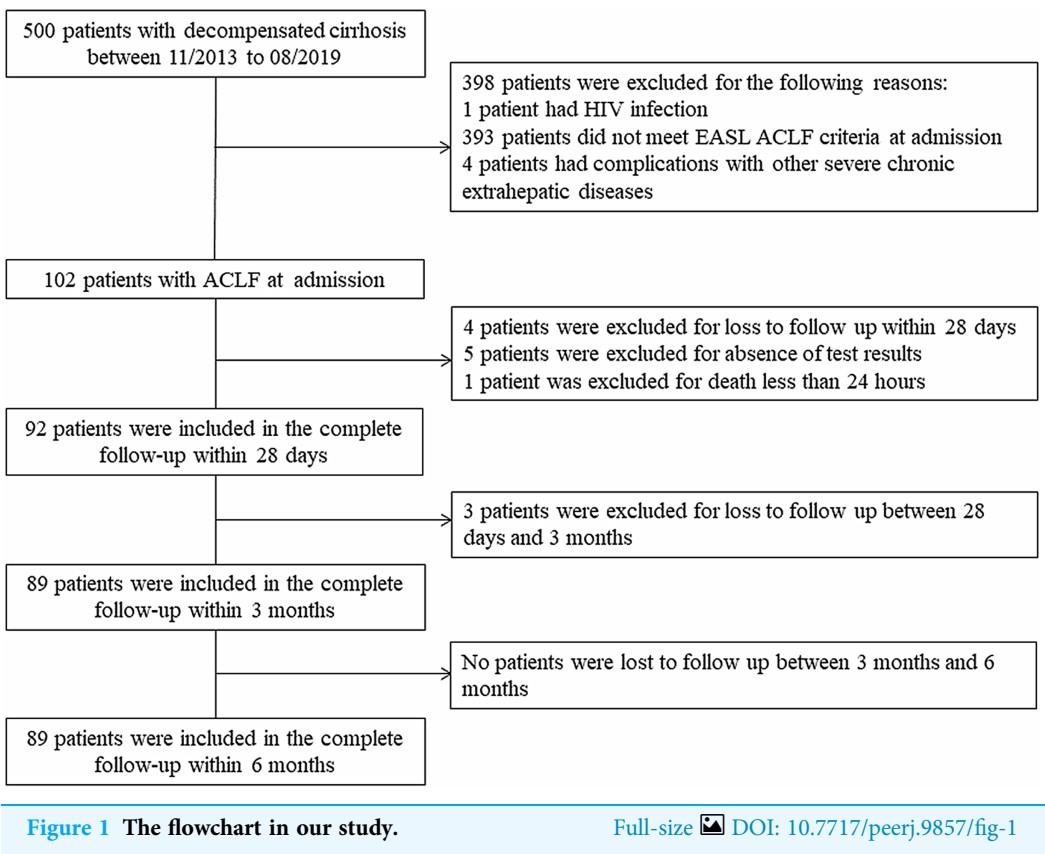

**Figure 1  The flowchart in our study.**               

prediction of 28-day, 3-month, and 6-month mortality in patients with ACLF. The specificity, sensitivity, negative likelihood ratio (NLV) and positive likelihood ratio (PLV) were computed for each cut-off value. The cut-off point was obtained by Youden's index with greatest Sensitivity and Specificity (*Youden, 1950*). The comparing of the areas under the ROC curve (AUROCs) was performed by Delong-test. 0.050 of two-tailed was significant meaning.

## RESULTS

### Characteristics of ACLF patients

There were 102 patients in this study. During the study period, 92 patients were enrolled in the analysis of the outcomes at 28 days; subsequently, 3 patients were lost to follow-up, and 89 patients were finally enrolled at both 3 and 6 months. The flowchart is shown in Fig. 1, and the demographic and biochemical characteristics of the study population are summarized in Table 1. The mean (±SD) age of the 102 patients was 56.96 (±12.18) years. The leading cause of decompensation events responsible for hospitalization was variceal bleeding (70/102, 68.6%). The ACLF patient distribution was grade 1 (31/102, 30.4%), grade 2 (45/102, 44.1%), and grade 3 (26/102, 25.5%). The most common degree of ascites was moderate (28/102, 27.5%), followed by severe (25/102, 24.5%) and mild (13/102, 12.7%). Forty-nine (48%) patients had undergone endoscopic hemostasis, 41 (40.2%) patients had undergone mechanical ventilation, and 66 (64.7%) patients had used

**Table 1 Characteristics of patients in the ACLF cohort.**

| | Patients with ACLF at admission (*n* = 102) | Patients in complete follow-up within 28-days (*n* = 92) | Patients in complete follow-up within 3-months or 6-months (*n* = 89) |
|---|---|---|---|
| Age, mean ± SD | 56.96 ± 12.18 | 57.5 ± 12 | 57.8 ± 12 |
| Sex (male), *n* (%) | 70 (68.6%) | 62 (67.4%) | 59 (66.3%) |
| Hospitalization days, median (IQR) | 4 (1–11) | 4.5 (1.25–11.0) | 5.0 (1.0–11.0) |
| Aetiology of chronic liver disease, *n* (%) | | | |
| Hepatitis B Virus | 59 (58.8%) | 52 (57.6%) | 50 (56.1%) |
| Alcoholic liver disease | 35 (34.1%) | 32 (34.7%) | 31 (34.8%) |
| Hepatitis C Virus | 2 (1.9%) | 2 (2.1%) | 2 (2.2%) |
| Primary biliary cirrhosis | 4 (3.9%) | 4 (4.3%) | 4 (4.5%) |
| Others | 17 (16.7%) | 15 (16.3%) | 15 (16.8%) |
| Primary reason for hospitalization, *n* (%) | | | |
| Variceal bleeding | 70 (68.6%) | 65 (70.7%) | 62 (69.6%) |
| Ascites | 6 (5.9%) | 5 (5.4%) | 0 (5.6%) |
| Hepatic encephalopathy | 14 (13.7%) | 13 (14.1%) | 13 (14.6%) |
| Infection | 11 (10.8%) | 8 (8.7%) | 8 (8.9%) |
| Others | 1 (0.9%) | 1 (1.1%) | 1 (1.1%) |
| ACLF grade, *n* (%) | | | |
| ACLF grade 1 | 31 (30.4%) | 29 (31.5%) | 28 (31.5%) |
| ACLF grade 2 | 45 (44.1%) | 39 (42.4%) | 37 (41.6%) |
| ACLF grade 3 | 26 (25.5%) | 24 (26.1%) | 24 (26.9%) |
| Endoscopic hemostasis, *n* (%) | 49 (48%) | 48 (52.2%) | 46 (51.7%) |
| The degree of ascites, *n* (%) | | | |
| Mild | 13 (12.7%) | 11 (12.0%) | 11 (12.3%) |
| Moderate | 28 (27.5%) | 27 (29.3%) | 25 (28.1%) |
| Severe | 25 (24.5%) | 24 (26.1%) | 24 (26.9%) |
| Hepatocellular carcinoma, *n* (%) | 10 (9.8%) | 10 (10.9%) | 9 (10.1%) |
| Mechanical ventilation, *n* (%) | 41 (40.2%) | 37 (40.2%) | 37 (41.6%) |
| Vasopressor use, *n* (%) | 66 (64.7%) | 60 (65.2%) | 58 (65.2%) |

Note:
ACLF, Acute-on-chronic liver failure; SD, Standard Deviation; IQR, interquartile range.

vasopressors. In the 28-day and 3-month analyses, the mean age was 57.5 (±12) years and 57.8 (±12) years, respectively, and 62 (67.4%) and 59 (66.3%) patients were male. The leading cause of liver cirrhosis is Hepatitis virus infection and variceal bleeding accounts for the majority of hospitalizations. The distributions of patients who were included in the complete follow-up within 28 days and were included in the complete follow-up within 3 months were similar to that of all 102 patients in terms of ascites grade, ACLF grade, and treatment strategy. A total of 47 (46.1%), 58 (56.9%), and 61 (59.8%) patients died within 28 days, 3 months, and 6 months, respectively. The causes of death at 6 months were as follows: 3 (4.9%) patients had cardiogenic shock, 6 (9.8%) patients

**Table 2 The comparison of prognostic scores.**

| Prognostic score | All patients (n = 92) | 28-Days | | | 3-Months | | | 6-Months | | |
|---|---|---|---|---|---|---|---|---|---|---|
| | | Survivors (n = 45) | Non-survivors (n = 47) | P-value | Survivors (n = 31) | Non-survivors (n = 58) | P-value | Survivors (n = 28) | Non-survivors (n = 61) | P-value |
| CTP score | 11 (9–13) | 10 (8–12) | 12 (10–14) | **0.001** | 10 (8–12) | 11.00 (10.00–13.25) | **0.028** | 10 (8–12) | 11 (10–13.5) | **0.033** |
| MELD score | 18 (14–25.75) | 16 (13.5–20) | 24 (15–29) | **0.004** | 15 (12–18) | 23 (15–29) | **0.001** | 15 (12–18) | 23 (15–29) | **0.002** |
| MELD-Na score | 20.69 (15.00–29.00) | 18.00 (14.00–27.36) | 24.00 (15.48–29.64) | 0.081 | 16.54 (13–26.13) | 23.27 (16–29.67) | **0.011** | 17.27 (14.00–24.73) | 23.00 (15.74–29.70) | **0.020** |
| CLIF-C OF score | 10 (9–11) | 9 (8–10) | 11 (10–12) | **<0.001** | 9 (8–10) | 10.00 (9.75–12.00) | **0.001** | 9 (8–10) | 10 (9–12) | **0.001** |
| CLIF-SOFA score | 10 (8–13) | 8 (6.5–10) | 12 (10–14) | **<0.001** | 8.55 ± 2.69 | 11.46 ± 3.36 | **<0.001** | 8.53 ± 2.67 | 11.33 ± 3.39 | **<0.001** |
| CLIF-C ACLF score | 49.59 ± 10.59 | 45.01 ± 9.99 | 53.98 ± 9.28 | **<0.001** | 44.39 ± 10.61 | 52.85 ± 9.41 | **<0.001** | 44.11 ± 10.36 | 52.56 ± 9.66 | **0.001** |

Notes:
CTP, Child-Turcotte-Pugh; MELD, model for end-stage liver disease; MELD-Na, model for end-stage liver disease-sodium; CLIF-C OF, chronic liver failure consortium organ function; CLIF-SOFA, chronic liver failure-sequential organ failure assessment; CLIF-C ACLF, chronic liver failure consortium acute-on-chronic liver failure.
$P$ value < 0.05 was considered significant and is indicated in bold.

had infectious shock, 12 (19.7%) patients had respiratory failure, 18 (29.5%) patients had hemorrhagic shock, 19 (31.1%) patients had liver-related complications (4 patients had liver failure, 15 patients had HE) and 3 (4.9%) patients had an uncertain cause of death. The causes of death at 28 days, 3 months, and 6 months are outlined in Table S1.

## Comparison of prognostic scores between the nonsurviving group and the surviving patients

The comparison of the six scores of patients with ACLF were shown in Table 2. ACLF patients were grouped into surviving and nonsurviving groups based on their 28-day, 3-month, and 6-month outcomes. The non-surviving patients had a higher CTP score, MELD score, CLIF-C OF score, CLIF-SOFA score and CLIF-ACLF score, compared with surviving patients ($P < 0.050$). Although the comparison of the MELD-Na score was not statistically significant ($P = 0.081$), it was still higher in the nonsurviving group. Statistically significant differences were found for the CTP score, MELD-Na score, MELD score, CLIF-SOFA score, CLIF-ACLF score, and CLIF-C OF score at 3 months and 6 months ($P < 0.050$).

## Predictive ability for 28-day, 3-month and 6-month outcome in ACLF patients

The discriminative ability of the CTP score, MELD score, MELD-Na score, CLIF-C OF score, and CLIF-ACLF score calculated for 28-day, 3-month, and 6-month survival is summarized in Table 3. At 28 days, the CLIF SOFA score had the highest AUROC (0.805, 95% CI [0.715–0.896]), followed by the CLIF-ACLF score (0.741, 95% CI [0.640–0.843]), CLIF-C OF score (0.712, 95% CI [0.676–0.869]), CTP score (0.707, 95% CI [0.600–0.813]), MELD score (0.673, 95% CI [0.560–0.787]), and MELD-Na score (0.606, 95% CI [0.487–0.724]). When predicting 3-month and 6-month mortality, the

**Table 3 The efficacy and performance comparison of the prognostic scores for predicting mortality in 28-day, 3-month and 6-month.**

| Prognostic score | ROC area (95% CI) | P-value | Cut-off point | Sensitivity (%) | Specificity (%) | PLV | NLV |
|---|---|---|---|---|---|---|---|
| 28-Days mortality | | | | | | | |
| CTP score | 0.707 [0.600–0.813] | <0.001 | 10.00 | 57.78 | 74.47 | 2.26 | 0.57 |
| MELD score | 0.673 [0.560–0.787] | <0.001 | 22.00 | 84.44 | 59.57 | 2.09 | 0.26 |
| MELD-Na score | 0.606 [1.487–0.724] | 0.006 | 22.00 | 71.11 | 59.57 | 1.76 | 0.48 |
| CLIF-C OF score | 0.712 [0.676–0.869] | <0.001 | 10.00 | 86.67 | 57.45 | 2.04 | 0.23 |
| CLIF-SOFA score | 0.805 [0.715–0.896] | <0.001 | 10.00 | 77.78 | 74.47 | 3.05 | 0.29 |
| CLIF-ACLF score | 0.741 [0.640–0.843] | <0.001 | 48.20 | 66.67 | 76.60 | 2.85 | 0.44 |
| CLIF-SOFA score vs CTP | 0.099 [0.019–0.179] | 0.017 | | | | | |
| CLIF-SOFA score vs MELD | 0.132 [0.025–0.240] | 0.016 | | | | | |
| CLIF-SOFA score vs MELD-Na | 0.200 [0.081–0.318] | 0.001 | | | | | |
| CLIF-SOFA score vs CLIF-C ACLF | 0.063 [0.009–0.164] | 0.038 | | | | | |
| CLIF-SOFA score vs CLIF-C OF | 0.054 [0.082–0.158] | 0.042 | | | | | |
| 3-Months mortality | | | | | | | |
| CTP score | 0.641 [0.521–0.760] | <0.001 | 12.00 | 90.32 | 36.21 | 1.41 | 0.27 |
| MELD score | 0.715 [0.598–0.832] | <0.001 | 19.00 | 80.65 | 62.07 | 2.13 | 0.31 |
| MELD-Na score | 0.664 [0.541–0.788] | <0.001 | 20.52 | 74.19 | 62.07 | 1.96 | 0.41 |
| CLIF-C OF score | 0.709 [0.595–0.822] | <0.001 | 9.00 | 64.52 | 75.86 | 2.67 | 0.47 |
| CLIF-SOFA score | 0.751 [0.646–0.857] | <0.001 | 10.00 | 80.65 | 67.24 | 2.46 | 0.29 |
| CLIF-ACLF score | 0.729 [0.615–0.842] | <0.001 | 48.20 | 74.19 | 74.14 | 2.87 | 0.35 |
| CLIF-SOFA score vs CTP | 0.111 [0.016–0.206] | 0.023 | | | | | |
| CLIF-SOFA score vs MELD | 0.037 [−0.069 to 0.141] | 0.396 | | | | | |
| CLIF-SOFA score vs MELD-Na | 0.089 [−0.023 to 0.207] | 0.126 | | | | | |
| CLIF-SOFA score vs CLIF-C ACLF | 0.043 [−0.019 to 0.106] | 0.109 | | | | | |
| CLIF-SOFA score vs CLIF-C OF | 0.023 [−0.037 to 0.113] | 0.420 | | | | | |
| 6-Months mortality | | | | | | | |
| CTP score | 0.640 [0.518–0.762] | <0.001 | 12.00 | 92.86 | 36.07 | 1.45 | 0.20 |
| MELD score | 0.708 [0.591–0.824] | <0.001 | 19.00 | 82.14 | 60.66 | 2.09 | 0.29 |
| MELD-Na score | 0.655 [0.532–0.777] | <0.001 | 20.52 | 75.00 | 60.66 | 1.91 | 0.41 |
| CLIF-C OF score | 0.716 [0.601–0.831] | <0.001 | 9.00 | 64.29 | 73.77 | 2.45 | 0.48 |
| CLIF-SOFA score | 0.742 [0.633–0.852] | <0.001 | 10.00 | 82.14 | 65.57 | 2.39 | 0.27 |
| CLIF-ACLF score | 0.725 [0.610–0.840] | <0.001 | 48.20 | 75.00 | 72.13 | 2.69 | 0.35 |
| CLIF-SOFA score vs CTP | 0.102 [0.001–0.205] | 0.042 | | | | | |
| CLIF-SOFA score vs MELD | 0.054 [−0.050 to 0.140] | 0.319 | | | | | |
| CLIF-SOFA score vs MELD-Na | 0.098 [−0.024 to 0.201] | 0.107 | | | | | |
| CLIF-SOFA score vs CLIF-C ACLF | 0.036 [−0.023 to 0.094] | 0.210 | | | | | |
| CLIF-SOFA score vs CLIF-C OF | 0.027 [−0.053 to 0.109] | 0.406 | | | | | |

Notes:
ROC, receiver operating characteristic; PLV, positive likelihood ratio; NLV, negative likelihood ratio; CTP, Child-Turcotte-Pugh; MELD, model for end-stage liver disease; MELD-Na, model for end-stage liver disease-sodium; CLIF-C OF, CLIF consortium organ function; CLIF-SOFA, chronic liver failure-sequential organ failure assessment; CLIF-C ACLF, CLIF consortium acute-on-chronic liver failure; CI, Confidence interval.
P value < 0.05 was considered significant and is indicated in bold.

CLIF-C SOFA score both had the highest AUROC (0.751, 95% CI [0.646–0.857]; 0.742, 95% CI [0.633–0.852], respectively), by contrast, CTP score both had the lowest AUROC (0.641, 95% CI [0.521–0.760]; 0.640, 95% CI [0.518–0.762], respectively). The ROC curves

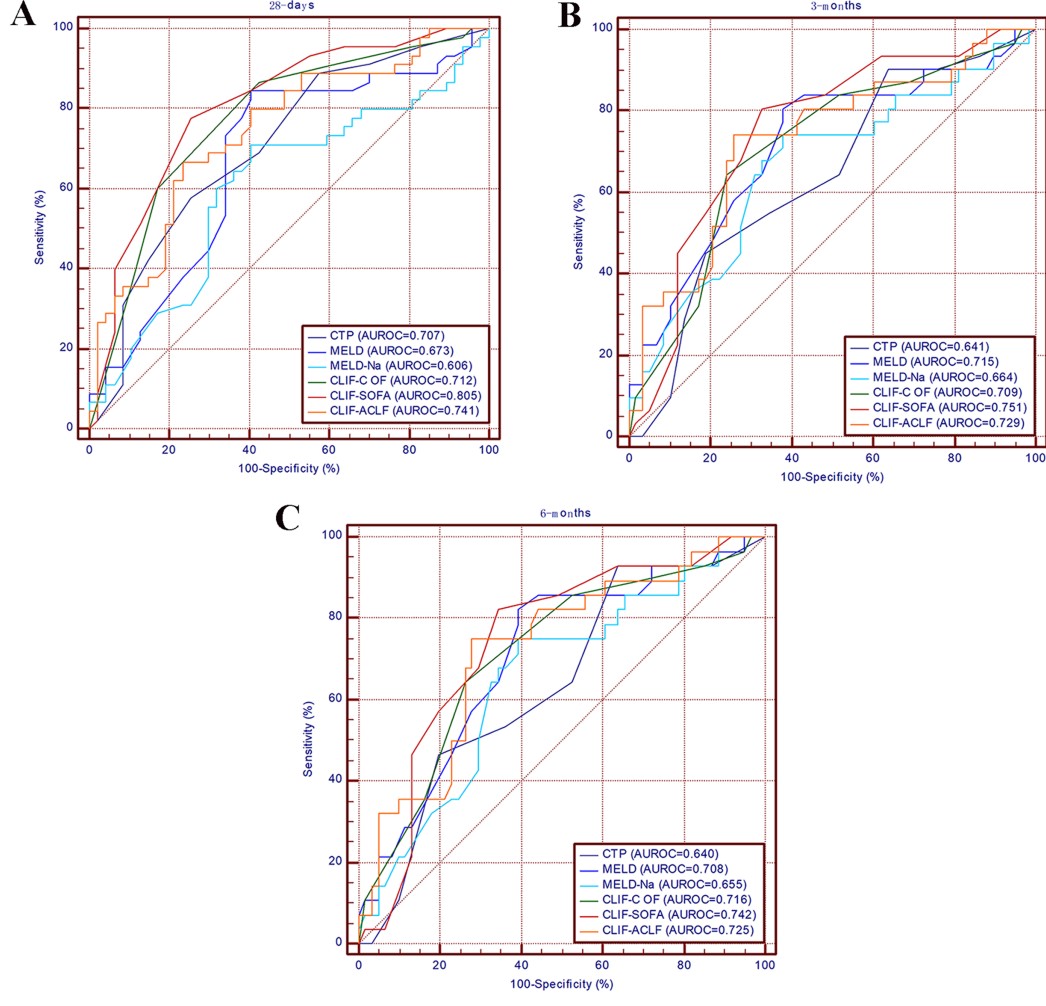

**Figure 2  Receiver operating characteristic curves for the MELD-Na score, MELD score, Child-Pugh score, CLIF-C OF score, CLIF-SOFA score and CLIF-ACLF score for predicting mortality at 28 days (A), 3 months (B) and 6 months (C).** MELD, the model for end-stage liver disease score; Child-Pugh, the Child-Pugh score.        

for the prognostic scores are shown in Fig. 2. All prognostic scores were able to predict mortality at 28 days, 3 months, and 6 months ($P < 0.050$).

## Comparing the predictive performance of all scores

As shown in Table 3, the AUROC of the CLIF-SOFA score is superior to those of the other five scores with regard to 28-day, 3-month, and 6-month mortality. The CLIF-SOFA has the highest predicting value in 28-day mortality with the AUROC of 0.805. The predicting performer of CLIF-SOFA is significantly higher than CTP score, MELD-Na score, MELD score, CLIF-C OF score, and CLIF-ACLF score ($P < 0.050$). At 3 months and 6 months, the comparison of AUROCs between the CTP score and the CLIF-SOFA score was statistically significant ($P < 0.050$); however, the comparisons of AUROCs between the CLIF-C OF score, CLIF-ACLF score, MELD-Na score and MELD

score were not significant ($P > 0.050$). At 28 days, the AUROC of MELD-Na was lower than other five scores.

## DISCUSSION

It is important to develop predictive scores that can identify patients who are at high risk of mortality, enabling the early provision of effective treatment to reduce mortality, especially in diseases with high mortality rates. ACLF is a clinical syndrome with a high mortality rate that is characterized by the development of acute decompensation (encephalopathy, ascites, gastrointestinal hemorrhage) and organ failure (such as kidney, renal, hepatic, coagulation, respiration, and circulation), so prognostic assessment is an indispensable for ACLF patients (*Moreau et al., 2013*). However, in the clinical setting, the prognosis is often hard to predict for certain patients because of different factors, such as etiology, disease stage, and complications. Previous studies have shown that many different scores have predictive value for mortality in ACLF patients. It is very important to choose the most efficient score for predicting mortality in Asian patients in clinical treatment. The clinical characteristics of ACLF patients in Asian is completely different form patients in Europe and America. In this study, the leading etiology of liver cirrhosis was hepatitis virus infection (58.8%), followed by alcohol-related cirrhosis (34.1%), which was similar to the primary etiologies of liver disease in most Asian countries.

It is not surprising that the mortality of ACLF patients was high in this study, as that it consistent with previous research (*Hernaez et al., 2019*; *Mahmud et al., 2019*; *Sundaram et al., 2019*). The mortality rate was 46.1% in the short term (28 days), and the mortality rate was 59.8% in the long term (6 months). The high mortality rate, which we find appalling, has spurred us to meaningfully contribute. Effective and inexpensive treatment strategies for patients with low socioeconomic status are limited because of the high costs associated with liver transplant and hemodialysis, partially in developing countries. The economical load produced by ACLF is still severe. Predicting the prognosis of patients with ACLF may be more important than treatment from the perspective of health economics for low-income families.

Recently, the CLIF-ACLF score, CLIF-C OF score, and CLIF-SOFA score have been used to evaluate prognosis in ACLF patients (*Engelmann et al., 2018*; *Song et al., 2018*). To the best of our knowledge, although the discriminative ability of these scores for predicting outcomes in ACLF patients has been illustrated, different conclusions have been drawn regarding the relative predictive value of these scores because of differences in study populations or observation durations.

The predictive value of the six scores (CTP score, MELD score, MELD-Na, CLIF-ACLF score, CLIF-C OF score, and CLIF-SOFA score) was compared at 28 days, 3 months, and 6 months. The AUROC of CLIF-SOFA is higher than other prognostic scores at 28 days, 3 months, and 6 months in our cohort, especially at 28 days. The CLIF-SOFA score provides a comprehensive and effective assessment of the severity of organ failure in ACLF patients and takes into account multiple systems, including the hepatic, renal, coagulation, respiratory, circulatory, and nervous systems; it was established by the European Liver Disease Collaboration Group for Liver Failure in 2013. *Sy et al. (2016)*

study indicated that the predictive value of the CLIF-SOFA score is better than those of the CTP score and MELD score for short-term outcomes. Any score has its advantages and disadvantages. Although the predictive value of the CLIF-SOFA score is high, the calculation of the CLIF-SOFA score is complicated due to the inclusion of more indicators. The Child–Pugh score is computed based on the PT, ascites, serum bilirubin, albumin, and HE (*Pugh et al., 1973*). The presence or absence of HE and ascites, which forms part of the CTP score, is subjective and has no clear cut-off value. The MELD score contains contains three indicators: the INR, creatinine, and bilirubin; it is vulnerable to confounding by hemorrhaging, ascites and the use of diuretics, with the absence of clearly defined cutoff values for categorizing cirrhotic patients (*Cholongitas et al., 2005*). The occurrence of hyponatremia is closely related to the prognosis of patients with cirrhosis, particularly patients with ascites; therefore, the MELD-Na score has been created based on the MELD score (*Biggins et al., 2006*). However, the MELD score had a lower AUROC than the other five scores at all time points in this study. This may be due to the main complications of patients in this study. The patients were mainly enrolled from the Department of Gastroenterology and needed endoscopic treatment for bleeding esophageal gastric varices (70/102, 68.6%). The number of cirrhosis patients with ascites as the primary reason for hospitalization was very small (6/102, 5.9%), Previous study have confirmed the ascites is the main complication of liver cirrhosis (*De Vusser et al., 1985*), and ascites is associated with a high risk of developing further complications of cirrhosis such as dilutional hyponatremia (*Piano, Tonon & Angeli, 2018*), Because of the number of patients with ascites are small, so the MELD-Na score may not play an important role in predicting patients' mortality, which may explain why the discriminative power of the MELD-Na score is lower than other five scores. The predicting value of the CTP, MELD-Na, and MELD scores in ACLF is not completely prefect because indicators reflecting systemic inflammation and organ failure is lacking. The CANONIC study had shown the advantage of the CLIF-ACLF, CLIF-SOFA, and CLIF-C OF scores over the CTP, MELD-Na, and MELD scores for the prediction of mortality in ACLF patients, which is according with the results in our study (*Jalan et al., 2014*). *Jalan et al. (2014)* first proposed the CLIF-C OF score in 2014 and proved that the value of the CLIF-C OF score is equivalent to that of the CLIF-SOFA score for the prediction of mortality. Considering the effects of WBC count and age on prognosis, *Jalan et al. (2014)* established the CLIF-ACLF score based on the CLIF-C OF score (*Hernaez et al., 2017*). The CLIF-ACLF score not only considers the effects of extrahepatic organ damage, coagulation and circulatory system failure on the prognosis but also includes the WBC count, which reflects the severity of inflammation; the CLIF-ACLF score was superior to the CTP, MELD-Na, and MELD scores (*Hernaez et al., 2017*). Despite the high predictive value of the CLIF-ACLF score and CLIF-C OF, these scores were established based on patients from European countries and the US with alcohol-related liver disease, and further researches are needed to explore whether they are applicable to Asian populations. Our research results have indicated that the scores also apply to Asian populations.

Several limitations existed in this study. First, this was a retrospective study, the number of patients included in our study was still not large, and some patients were lost to follow-up, which may have resulted in selection bias. Second, the scores were evaluated when admission to hospital and did not reflect the dynamic changes. Finally, the leading etiologies in patients in our study were hepatitis B virus infection, but most of the patients were diagnosed according to the EASL-ACLF criteria, leading to etiological bias.

In conclusion, our data reveal that the CTP score, MELD score, MELD-Na, CLIF-C OF score, CLIF-SOFA score, and CLIF-ACLF score are effective tools for predicting the prognosis in ACLF patients. The CLIF-SOFA score has better discriminative power for the evaluation of short-term mortality, and may help improve the management of ACLF patients.

### Funding
This study was supported by the National Natural Science Foundation of China (grant number: 81660110) and the "Gan-Po Talent 555" Project of Jiangxi Province. The funders had no role in study design, data collection and analysis, decision to publish, or preparation of the manuscript.

### Grant Disclosures
The following grant information was disclosed by the authors:
National Natural Science Foundation of China: 81660110.
"Gan-Po Talent 555" Project of Jiangxi Province.

### Competing Interests
The authors declare that they have no competing interests.

### Author Contributions

- Yue Zhang conceived and designed the experiments, analyzed the data, prepared figures and/or tables, and approved the final draft.
- Yuan Nie conceived and designed the experiments, performed the experiments, prepared figures and/or tables, authored or reviewed drafts of the paper, and approved the final draft.
- Linxiang Liu performed the experiments, analyzed the data, authored or reviewed drafts of the paper, and approved the final draft.
- Xuan Zhu analyzed the data, authored or reviewed drafts of the paper, and approved the final draft.

### Human Ethics
The following information was supplied relating to ethical approvals (i.e., approving body and any reference numbers):

The Ethics Committee of The First Affiliated Hospital of Nanchang University (2015-1203).

## Data Availability

The data is available in the Supplemental Files.

## Supplemental Information

Supplemental information for this article can be found online at http://dx.doi.org/10.7717/peerj.9857#supplemental-information.

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
