# Peer review of "Assessing the prognostic scores for the prediction of the mortality of patients with acute-on-chronic liver failure: a retrospective study"

_PeerJ, doi:10.7717/peerj.9857_

## Round 0.1 · original submission · Major Revisions

Thank you for resubmitting this interesting work. As the reviewers suggest it would be better to add validation results and please address all of the comments proposed by the reviewers in a point-by-point response.

·

Basic reporting

The article is well-written in English language, although there are some minor erratum along the text that should be corrected in a further revised version.

Experimental design

The study is well designed and address the hypothesis. However, I think it would be interesting to perform an additional analysis to find the optimal cut-off point of each ROC curve.

In table 3 authors already present a cut-off point, but it is not mentioned along the manuscript if this is the optimal cut-off point based on the Youden's index. If so, authors should mention it in the methodology section. If not, I would recommend to calculate the "optimal cut-off point" where both sensitivity and specificity maximize using the above mentioned Youden index.

Validity of the findings

No comment.

Additional comments

This is an interesting study that compares the predictive and discriminative power of several scores in acute-on-chronic liver failure patients.

In addition to my previous comments (correct some minor erratum throughout the manuscript and calculate the optimal cut-off point for each ROC curve), I would recommend authors to summarize the main findings in the section "Predictive ability for 28-day, 3-month and 6-month mortality in ACLF patients." As authors state, these findings are shown in Table 3, so I recommend to write only the main findings in order to facilitate the reading.

Despite these minor issues, I think this is a great and interesting work.

·

Basic reporting

No comment

Experimental design

- line 179,180 and table, liver failure, and encephalopathy were mentioned separately as causes of death at 6 months. You can possibly combine them as liver-related mortality and subgroup causes.
- Do you have reasons for loss of follow up?

Validity of the findings

-line 272, you mentioned patients required treatment for bleeding oesophageal and gastric varices. Do you have data on the type of variceal bleed and treatment modality?
- For a single-centre study, the conclusion is too strong to generalize results to whole China.

Reviewer 3 ·

Basic reporting

No comment

Experimental design

No comment

Validity of the findings

No comment

Additional comments

The authors present a retrospective study for assessing the prognostic scores for the prediction of the mortality of patients with acute-on-chronic liver failure. There are major points that need to be addressed:
1. It is important to have a validation cohort to validate the performance.
2. The results are written unclear. For example, the authors don't need to write any AUC values for every scores. They could be explained by only a table or figure and with some text description.
3. How did the authors define normal distribution and abnormal distribution?
4. ROC curve and AUC had been used in previous works in biomedical such as PMID: 31921391, PMID: 31277574, and https://doi.org/10.1016/j.neucom.2019.09.070. Therefore, the authors are suggested to refer more works in this description.
5. It is important to have some comparisons to the previous works on the same problem.
6. The ROC curves should be presented together with AUC values (i.e. Fig. 2)

---

## Round 0.2 · Minor Revisions

We appreciated your efforts to significantly improve the manuscript; however before we consider the acceptance these issues proposed by the reviewers should be solved point-by-point. We are looking forward to your revision.

·

Basic reporting

The article is well-written, and authors have made an effort and corrected the erratum detected in the prior version.

However, there are some minor issues to be corrected:

Line 110: "The MELD: 3.8×log (bilirubin) +9.6×log(creatinine) +11.2×log (INR)+6.43 (reference 8)."

It seems this sentence is incomplete. I would rewrite it as "The MELD formula was: ..."

Line 111: please, correct "below" instead of "bellow"

Line 147: There are 102 patients in this study.

I think this sentence should be in past tense: "There were 102 patients in this study."

Line 271: "The predicting value of the CTP, MELD-Na, and MELD scores in ACLF is not complete prefect because indicators reflecting systemic inflammation and organ failure is lacking."

I would suggest: "The predicting value of the CTP, MELD-Na, and MELD scores in ACLF is not completely perfect because indicators reflecting systemic inflammation and organ failure is lacking."

Nonetheless, as aforementioned, the article is well-written and references are relevant. Tables and figures summarize the main findings and are completely understandable.

Experimental design

The authors have included the information about the Youden index in the methods section, as requested. However, I would suggest to reference this work:

Youden, W.J. (1950). «Index for rating diagnostic tests». Cancer 3: 32-35 (PMID:15405679)

Apart from this suggestion, I have no further comments. The study question is well defined and addressed, and methodology is sufficiently explained.

Validity of the findings

I have no suggestions nor comments regarding the validity of findings. Conclusions are well defined and answer the research question based on the findings, and the clinical relevance of these findings is notable.

---

## Round 0.3 · accepted · Accept

Thank you for the revision, and I thought the comments by reviewers and editors are fully addressed.